# The Cytoskeleton and Its Roles in Self-Organization Phenomena: Insights from *Xenopus* Egg Extracts

**DOI:** 10.3390/cells10092197

**Published:** 2021-08-26

**Authors:** Zachary M. Geisterfer, Gabriel Guilloux, Jesse C. Gatlin, Romain Gibeaux

**Affiliations:** 1Department of Molecular Biology, University of Wyoming, 1000 E. University Ave, Laramie, WY 82070, USA; zgeister@uwyo.edu; 2Univ Rennes, CNRS, IGDR (Institut de Génétique et Développement de Rennes)-UMR 6290, F-35000 Rennes, France; gabriel.guilloux@univ-rennes1.fr

**Keywords:** cytoskeleton, *Xenopus* egg extract, self-organization, microtubule, actin

## Abstract

Self-organization of and by the cytoskeleton is central to the biology of the cell. Since their introduction in the early 1980s, cytoplasmic extracts derived from the eggs of the African clawed-frog, *Xenopus laevis*, have flourished as a major experimental system to study the various facets of cytoskeleton-dependent self-organization. Over the years, the many investigations that have used these extracts uniquely benefited from their simplified cell cycle, large experimental volumes, biochemical tractability and cell-free nature. Here, we review the contributions of egg extracts to our understanding of the cytoplasmic aspects of self-organization by the microtubule and the actomyosin cytoskeletons as well as the importance of cytoskeletal filaments in organizing nuclear structure and function.

## 1. Introduction

The use of cell-free Amphibian egg extracts was first described by Lohka and Masui, who added demembranated sperm nuclei of *Xenopus* into cytoplasm from eggs of *Rana pipiens* to investigate events such as nuclear envelope formation and the initiation of DNA synthesis [1]. In subsequent work, Lohka and Maller took advantage of the well-established husbandry of a different frog species and pioneered the development of extracts derived from the eggs of the African clawed-frog, *Xenopus laevis*. Incubating isolated sperm nuclei in extract resulted in the formation of spherical interphase nuclei, which could be further induced through hallmark stages of mitosis, including nuclear envelope breakdown (NEBD) and spindle formation by simply adding an equal volume of egg extract prepared in the presence of a Ca^2+^ chelating agent [2,3]. Soon after, Andrew Murray and colleagues further optimized extract preparation to allow for multiple cell cycles in vitro, as well as induction of CSF (Cytostatic Factor) extracts into interphase [4,5,6]. In the years that followed, *Xenopus* egg extracts were used to study multiple events of the cell cycle, including DNA replication [7], vesicle fusion [8], cyclin B degradation [9], and microtubule dynamics as well as spindle assembly [10,11,12]. *Xenopus* egg extract would subsequently flourish as a major model system thanks to its simplified cell cycle, large experimental volumes, biochemical tractability and accessible cell-free nature. With advances in proteomics, the concentrations of the most abundant ~11,000 cytoplasmic proteins are known [13], making it an excellent system for interrogation of cell physiology and cytoskeletal dynamics.

In metazoan cells, the three cytoskeletal elements, microtubules, F-actin and intermediate filaments, act to execute a variety of complex cellular phenomena, including cell motility, cell division and cytoplasmic organization, by active and cooperative self-organization. Cytoskeletal self-organization, specifically in the case of microtubules and F-actin, relies fundamentally on the fact that these dynamic polar filaments self-assemble from the non-covalent binding of globular monomers, with a higher level of complexity emerging from the self-organization of those filaments into various three-dimensional assemblies. This structural organization involves other components, including regulators of filament nucleation and dynamics, as well as molecular motors [14,15]. One mechanism of organization is the directional transport of filaments using forces generated by molecular motors which bind and crosslink cytoskeletal filaments [16]. Self-organization is also affected by confinement [17], gravitation, gradients of concentrations and interactions between different types of cytoskeletal fibers [18]. Adding to the complexity of this emergent phenomenon is the spatial and temporal control of filament nucleation and dynamics, which in turn is influenced by interplay with other sub-cellular structures and organelles within the cytoplasm. All of these facets of cytoskeleton-dependent self-organization have been investigated using *Xenopus* egg extract (Figure 1).

Although *Xenopus* egg extract systems have been instrumental in the study of mitotic spindle assembly, and more recently in facilitating interspecies comparisons to better understand mechanisms of spindle size regulation (reviewed in [19]), they have also been broadly used to investigate microtubules at all stages of the cell cycle. Such studies include investigation of interphase aster assembly and dynamics [20,21,22], as well as the aster’s role in cell-like compartmentalization [23] and the positioning of both organelles [24] and the cytokinesis machinery [21]. Furthermore, egg extract studies have enabled a better understanding of the role of microtubules in nuclear disassembly [25] and nuclear size regulation [26,27]. In the case of F-actin, work in *Xenopus* egg extract has shown it to be involved in chromosome alignment and in nuclear assembly [28] as well as serving an indirect role in DNA replication [29]. Prior to 2011, the study of F-actin’s cytoplasmic roles in extract was largely precluded because until this point in time, most extract preparations utilized cytochalasin to inhibit actin polymerization and prevent cytoplasmic gelation (a notable exception includes work from the Bement lab [30]). In a breakthrough study, Field and colleagues showed that bulk contraction driven by actomyosin was cell cycle regulated and specific to metaphase extract [31]. Cytochalasin-free actin-intact egg extract was subsequently used to study the interplay between microtubule and actomyosin networks [21,23,24]. Such an extract was also utilized in combination with confinement to investigate the flow of F-actin filaments [32], actomyosin-based formation of intracellular aggregates [33] and boundary-dependent generation of forces [34,35]. The last major cytoskeletal system includes intermediate filaments, which are composed of six protein sub-types, including the nuclear lamins, which have been extensively studied in *Xenopus* egg extracts, resulting in many key discoveries about the role of lamins in nuclear assembly and in the regulation of many nuclear processes. In fact, egg extract allowed differentiation of the three types of nuclear lamins [36,37,38] and facilitated the study of how they contribute to the formation of the lamina [39] and the assembly of nuclear envelopes containing pores [40,41]. Furthermore, nuclear lamina has also been shown to be involved not only in DNA replication [42,43], but also in post-mitotic chromatin decondensation [44] and phosphorylation-mediated regulation of nuclear size [45,46]. We should note that several studies have also partially characterized the keratin cytoskeleton in extracts; while it is clear that it interacts with other cytoskeletal systems, the body of experimental data specifically from *Xenopus* egg extracts is limited [47,48].

Over the past 35 years, *Xenopus* egg extracts have proven to be an invaluable system to study the cytoskeleton and have provided new insight into some of the key roles in self-organization phenomena. In this review, we focus on the use of *Xenopus* egg extract to study the cytoskeletal organization beyond that of the mitotic spindle. We first describe the cytoplasmic aspects of self-organization by the microtubule and the actomyosin cytoskeletons. We then turn towards the nucleus and address the importance of all three types of cytoskeletal filaments in organizing nuclear structure and function.

## 2. Recapitulating Interphase Aster Growth and Expansion in *Xenopus* Egg Extracts

Most early studies of the microtubule cytoskeleton in *Xenopus* egg extracts were focused on mitotic spindle assembly, which was induced by simply adding an equal volume of CSF-arrested meiotic extract to interphase extract containing nuclei. It was not until the advent of artificial microtubule organizing centers (aMTOCs) that studies of the interphase aster became more experimentally tractable. Interphase microtubule asters consist of branched radial arrays of microtubules focused around an MTOC, typically the centrosome in animal cells or the spindle pole body (SPB) in yeast cells. The experimental underpinnings for this advance came initially from investigations into the roles of the mitotic kinase Aurora A (AurA). Tsai and Zheng originally set out to deplete AurA from *Xenopus* egg extracts using antibodies against the protein coupled to magnetic beads as a means to determine whether the kinase played a role in mitotic spindle formation. Interestingly, anti-AurA beads left in the egg extract after immunodepletion were observed to localize at aster centers and spindle poles. In the presence of RanGTP and M phase egg extract, these AurA beads were capable of nucleating microtubules, earning them the moniker of aMTOCs [49]. When added to interphase *Xenopus* egg extracts, these aMTOCs generate symmetric, radial asters that are similar to those observed in *X. laevis* zygotes [20,21,50].

Historically, it was thought that most, if not all interphase microtubules, were generated with their minus-ends anchored at the centrosome and their plus-ends growing radially outward, with many extending all the way to the cell cortex [51]. In the case of extremely large cells, such as those of the early blastomere in *X. laevis*, this traditional view of aster architecture would require that individual microtubules grow to achieve lengths greater than 500 µm and it would not account for observed microtubule densities in the aster, which remain relatively constant as a function of distance from the aster center. Indeed, astral microtubules are thought to be an order of magnitude smaller than the largest dimension of the aster itself, (~5–16 μm [20,52,53]), suggesting that interphase asters must rely on mechanisms other than simple microtubule growth to occupy large cytoplasms. By supplementing *Xenopus* egg extracts with AurA beads, Ishihara et al. were able to form interphase asters and characterize the plus-end density of microtubules as a function of distance from the aMTOC. These measurements revealed that microtubule plus-end density remains constant, or in some cases, increases as a function of distance from the aMTOC [20]. These observations challenged the radial elongation model, where plus-end density would be expected to decrease as a function of distance from the organizing center. Instead, Ishihara et al. suggested an alternate source of microtubule nucleation (distal from the centrosome) might be contributing to aster growth and ultimately its ability to scale with changes in cell size.

Indeed, nucleation of microtubules from noncentrosomal locations had been observed before (reviewed in [54]), however the specific identity of acentrosomal nucleation sources that contribute to the microtubule mass of the interphase aster in *Xenopus* egg extracts remain unknown. One such source might be microtubule-dependent microtubule nucleation observed in *Xenopus* egg extracts by Petry et al., who incubated template microtubules in meiotic *Xenopus* egg extracts supplemented with a constitutively active RanGTP (RanQ69L) [53]. Here, the generation of a new microtubule is thought to rely primarily on the activation of augmin by TPX2, and the subsequent recruitment of ɣ-tubulin to the side of a pre-existing microtubule [55,56,57,58]. However, whether this type of nucleation occurs specifically in the interphase cytoplasm, which is thought to be largely devoid of TPX2 due to the protein’s partitioning to the nucleus [59,60] and in which the GDP-bound form of Ran predominates, remains to be determined. In addition to microtubule-dependent microtubule nucleation, microtubule severing enzymes have also been implicated in generating microtubule mass distant from the organizing center (reviewed in [61]), but it has yet to be established whether this mechanism plays a direct or indirect role in the establishment and maintenance of the interphase aster microtubule architecture.

Since both the nucleation of microtubules and the growth of pre-existing microtubules vie for the same-shared soluble tubulin pool, it is likely that these two facets of aster growth provide reciprocal feedback on one another to determine the steady-state size of the aster. Studies in confined volumes of *Xenopus* egg extracts measured a negative relationship between the number and local density of growing microtubule ends and the polymerization rate of the microtubules, suggesting that certain growth promoting molecules may be limited at higher microtubule plus-end or polymer densities [22]. Similarly, microtubule nucleation may also be regulated in *Xenopus* egg extracts by the spatial abundance of nucleation factors and or catastrophe factors [62]. New models describing aster growth and expansion will need to account for both the polymerization dynamics of individual microtubules as well as the nucleation of additional microtubules from sources other than the centrosome [20,53,63].

## 3. Aster Centration and Cell-Like Compartmentalization in *Xenopus* Egg Extracts

Critical to the spatial organization of the eukaryotic cytoplasm is the proper positioning of the interphase aster. Typically, the center of the aster positions itself at the geometric center of the cell, where it is able to coordinate anterograde and retrograde vesicular trafficking and serve as a scaffold to spatially organize organelles. In most cases, the aster center is often physically associated with the surface of the nucleus and thus, its position at the onset of mitosis and NEBD determines where spindle assembly occurs, and ultimately the position of the cleavage furrow [61,64,65]. The orchestration of aster centration and nuclear movement is perhaps most striking during pronuclear migration in large blastomeres, where the male pronucleus and the associated centrosomal aster must migrate over long distances through a viscous cytoplasm to reach the center of the cell [65], a process that inherently requires the generation of force. Though this phenomenon has been extensively studied in early blastomeres of sea urchin and *Xenopus* embryos, *Xenopus* egg extracts have only recently been used to investigate how asters generate (or respond to) forces within the cell. Sulerud and colleagues entrapped individual aMTOCs in *Xenopus* egg extracts enclosed within hydrogel “micro-enclosures” of different geometries and visualized their movement toward the enclosure’s geometric center [66]. Importantly, this centration was unaffected by perturbation of cytoplasmic dynein function. This observation undermined the possible contribution of motor-dependent pulling mechanisms, instead validating a mechanism that relied on microtubule-based pushing forces and one that had been previously observed in vivo [67]. The authors posited that microtubules pushed the aMTOCs away from proximal barriers by polymerizing against them, ultimately resulting in aster centration.

An emergent property of the expanding, centering aster is the subsequent organization of the cytoplasm. In bulk *Xenopus* egg extracts supplemented with multiple AurA beads, aMTOC seeded asters expand until reaching the leading edge of microtubules originating from nearby asters, resulting in tessellated expanses of evenly spaced asters that were relatively homogenous in size [20,21,68]. Interestingly, zones of reduced microtubule density formed between juxtaposed asters where many proteins involved in cytokinesis, such as the chromosomal passenger complex, are recruited. This suggested that interactions of antiparallel microtubules promote cleavage furrow localization and self-organization [21]. These results were consistent with observations of cleavage furrow induction by antiparallel astral microtubule arrays in sand dollar blastomeres in seminal studies by Rappaport and colleagues [69]. More recently, Cheng and Ferrell supplemented *Xenopus* egg extract with demembranated sperm nuclei and fluorescent probes for DNA, microtubules, mitochondria and the ER and observed, over time, interphase-like cellular organization in a MT-dependent manner [23]. Notably, these cell-like compartments were able to undergo multiple rounds of division, suggesting that the interphase aster is capable of organizing the cytoplasm in a manner that allows recapitulation of cell division, even without a cell membrane. Similar studies investigated the interaction between growing microtubule asters and the cytoplasm of *Xenopus* egg extract, in which co-movement of asters with ER, mitochondria and other organelles was also observed [24]. These observations are consistent with a mechanically integrated cytoplasm that relies on the microtubule cytoskeleton and cytoplasmic dynein for organelle movement and positioning (Figure 2).

## 4. Studies of the Actomyosin Cytoskeleton in Bulk *Xenopus* Egg Extracts

The dynamics of actomyosin cytoskeleton in bulk cytoplasm had been extensively studied in extracts derived from tissue culture cells (e.g., [70,71,72]). The eventual establishment of cell-free extracts derived from *X. laevis* eggs [1] offered the same experimental advantages as tissue culture cell-derived extracts but provided additional tractability and ease-of-use, as it enabled exquisite control of the cell cycle as well as the ability to produce a large amount of extract without requiring large-scale cell production (and associated time and expense). Moreover, the larger size of *X. laevis* eggs (relative to typical somatic cells) results in less dilution during extract preparation, producing a more physiologically representative system that presumably more accurately recapitulates the cellular events being studied.

In pioneering studies, investigators visualized both MT and actomyosin in *Xenopus* egg extracts to better understand the interplay between the two systems and established that they do indeed physically interact via cytosolic crosslinkers and that this interaction allowed one to influence the other in terms of spatial organization and growth dynamics [30]. More recently, Field et al. used bulk *Xenopus* egg extracts to characterize cell cycle-dependent changes in the dynamics and behavior of actomyosin networks [31]. This study was motivated in part by previous observations of cell cycle-dependent differences in the mechanical properties of *X. laevis* oocyte cytoplasm [73]—the cytoplasm of M-phase oocytes was found to be significantly stiffer and more cohesive than interphase cytoplasm. Importantly, these differences were not dependent on an intact microtubule cytoskeleton, implicating an alternative mechanism and focusing subsequent studies on other cytoskeletal systems. Indeed, biochemical characterization of crushed eggs showed that actin and myosin were highly enriched in the condensed and refractive aggregates that appeared in cytoplasm isolated from individual meiotic oocytes [74]. Though incredibly informative, these early studies did not address the mechanistic basis for these actomyosin-dependent changes nor their cell cycle-dependent dynamics.

To address this gap in collective knowledge, Field et al. combined *Xenopus* egg extracts with time-lapse dark-field and fluorescence microscopy to visualize actomyosin networks in bulk extract volumes [31]. In mitotic (M-phase) extracts, they observed the formation of dense, F-actin rich condensates which grew in size over time, and seemed to result from the accumulation of sieved cytoplasmic components brought to the aggregate by periodic and contractile actomyosin waves, a phenomenon the authors termed “gelation-contraction” (Figure 3). This process was found to be more pronounced during mitosis, as untreated interphase (I-phase) extracts showed little or no contraction. This result was consistent with characterization of cell-cycle differences in the F-actin cytoskeleton—F-actin in M-phase extracts contained robust actin bundles that appeared to be organized in a gel. In contrast, the I-phase F-actin network was sparser, less bundled and appeared less interconnected. Molecular characterization of the mechanism for the cell cycle dependence suggested that Arp2/3, the F-actin associated nucleator, is more active during mitosis and that this increased activity might produce a sufficient density of F-actin filaments to promote the formation of an interconnected gel, an implicit requirement of the contraction observed. The authors presented a model in which periodicity resulted from a cycle of F-actin growth, gelation, inward contraction and subsequent diffusion of F-actin nucleators from the actomyosin gel, but could only speculate about the biophysical basis for the periodic nature of contraction. Recent studies of actomyosin behavior in bulk extract have not been focused specifically on the actomyosin per se, but instead tangentially explore the role that the network plays in movement and centration of MT asters [24] and in cellular compartmentalization ([23]; two papers discussed elsewhere in this review).

## 5. Studies of the Actomyosin Cytoskeleton in Discrete Volumes of *Xenopus* Egg Extracts

There is some real experimental utility in being able to control cytoplasmic volume and shape—it allows for simplification of complex cellular geometries and thus enables more predictive and testable biophysical models of cellular phenomena. As one might imagine (or perhaps knows through experience), it is difficult to manipulate the size and shape of individual cells (e.g., using micro-patterned surfaces [75] or PDMS (polymethylsiloxane) impressions [76]). However, in contrast, it is relatively straightforward to manipulate the size and shape of *Xenopus* egg extract simply because it is a liquid and takes the shape of whatever container you put it in. The open nature of *Xenopus* egg extracts, which makes it amenable to biochemical manipulations, can still be exploited before the extract is re-confined, allowing for addition of fluorescent proteins and/or dyes to facilitate visualization of the cytoskeleton and other cytoplasmic components via epifluorescent light microscopy.

Recent studies have exploited these advantages to investigate the effects of confinement on cytoskeletal systems by either mixing extract and oil to generate extract-oil emulsions with tunable droplet sizes [32,77,78,79,80,81] or by confining extract volumes in small photopatterned enclosures [22,66]. Pinot et al. used extract-in-oil emulsions to investigate whether F-actin dynamics and organization are influenced by confinement [32]. F-actin flow was observed in droplets and measured by tracking labeled F-actin filaments moving through the extract. This flow had hallmarks of gelation-contraction observed previously in bulk extracts in that it could generate a single, heterogeneous aggregate of refractive particles (as observed using brightfield microscopy) and the flow was typically directed inward toward the droplet center. However, in contrast to actomyosin behavior observed previously in bulk extracts, the observed F-actin flow seemed to be faster under confinement and did not exhibit a cell cycle dependence since it occurred in both M-phase and I-phase, though the velocity was slightly reduced in I-phase extract. Inhibition of myosin did not completely abolish F-actin flow and resulted in only a small decrease in F-actin flow rates. Thus, in these initial studies, F-actin flows induced by confinement seemed to possess different mechanistic underpinnings than gelation-contraction in bulk extract. Though no substantive explanation was provided for these conflicting results, it is possible that experimental differences in F-actin labeling (GFP-Lifeact vs. phalloidin) and differences in extract preparation might have contributed.

In addition to F-actin flow, confinement of extract has been shown to elicit the emergence of distinct actomyosin-based structures. The endogenous actin nucleation activity of extracts supplemented with GFP-Lifeact and an ATP regeneration system is capable of producing steady-state contracting actomyosin networks, characterized by a central occlusion that excludes F-actin (likely an organellar aggregate akin to that observed in bulk extracts) at the droplet’s geometric center. This occlusion is surrounded by a radially symmetric network of F-actin with a high density at the occlusion boundary that decreases monotonically as the network extends toward the droplet periphery [33]. Interestingly, these actomyosin structures exhibit a dynamic steady-state flow directed toward the occlusion center yet somehow maintain a stationary F-actin density gradient. The ability to generate this invariant and persistent network enabled further characterization of the intrinsic flows. Surprisingly, the uniform rate of contraction in the system was found to be independent of F-actin density and of the geometry of the confinement. Certain molecular perturbations of F-actin assembly and disassembly were found to have a pronounced impact on both the extent and flow of the networks. However, if the ratio between the rate of local F-actin turnover and that of network contraction remained constant, so did the architecture of the resulting network. Only conditions that perturbed this ratio led to dramatic changes in network organization and dynamics. In summary, these studies indicate that the actomyosin system is both robust and tunable, and provide unique insight as to the underlying biophysics that might be applicable to other contractile actomyosin networks, e.g., the cytokinetic ring.

The stereotypical positioning of certain organelles within cells, namely the nucleus and mitotic spindle, has long intrigued cell biologists. Most studies aimed at elucidating the biomechanics of this centration have been focused on the microtubule cytoskeleton (see section on microtubule studies in extract), which is not surprising given that the interphase aster directly interacts with the cell nucleus and the mitotic spindle is itself composed mostly of microtubules. However, the observation that actomyosin networks could center themselves in confined extracts was incredibly interesting in that it hinted at a microtubule-independent centering mechanism. Indeed, addition of nocodazole to extracts prior to droplet encapsulation had no effect on the positioning of actomyosin aggregates within the droplets, suggesting that the mechanism did not depend on an intact microtubule cytoskeleton. Conversely, the structure of actin networks affects both microtubule dynamics and the motility of microtubule asters reconstituted within encapsulated extracts [80].

These extract-in-oil emulsions were used to study how the size of confinement affected the ability of the network to center [34,35]. Both groups independently identified an abrupt shift in aggregate localization from the edge of the droplet (decentered) to centered aggregates as the size of the confining droplet was increased. Centering was thought to result from the inward directed flow of the actomyosin network that surrounded the system. Ierushalmi et al. observed a continuous steady-state flow of the network directed toward the aggregate center [33], much like that described previously. A hydrodynamic biophysical model that took into account the properties and flows of the cytosol was used to accurately describe this centering behavior. To describe observed decentering, the authors evoked a mechanism whereby a stochastic interaction with “cortical” actin at the inner surface of the droplet boundary could generate a pulling force to move the aggregate to the droplet edge, with the probability of such an interaction occurring increasing as the droplet size was decreased. In contrast to a steady actomyosin flow toward the aggregate center, Sakamoto et al. observed contractile waves in their droplet studies [34,35], similar to that observed previously in bulk extract [31]. By modulating the inner surface properties of the extract droplets, the authors were able to shift the propensity of the system to decenter, leading them to generate a theoretical model that relied on the antagonism between decentering forces generated by actomyosin bridges spanning from the aggregate center to the droplet edge and centering forces generated by actomyosin contraction. In this model, the final state of the system was dependent on the relative dynamics of bridge maturation and a droplet-size dependent change in contraction wave frequency. Though there were some notable experimental differences in the two approaches, e.g., Sakamoto et al. used squashed droplets whereas Ierushalmi and colleagues conducted experiments in spherical droplets, it remains unclear why the dynamics of actomyosin contraction differed between each set of experiments. Regardless, the fact that the system can center itself under the right conditions suggests that this mechanism might contribute to in vivo centering phenomena.

## 6. Importance of Nuclear Lamina Assembly for Nuclear Processes and Envelope Integrity

An important nuclear cytoskeletal element, and undoubtedly the most studied one, is the lamina. Described in early electron microscopy studies in amoebae [82] and in parasitic gregarines of grasshoppers [83] as a supporting layer of fine filaments at the inner side of the nuclear envelope, the lamina is today known as type V intermediate nuclear filaments composed of A-, B- and C-type lamins [84]. Using *Xenopus* egg extracts, many key discoveries have been made about the roles of lamins in nuclear assembly and in the regulation of many nuclear processes (Figure 4, (1)–(2)).

In pioneering work using *Xenopus* nuclear reconstitution extracts prepared from calcium inophore-treated eggs, Newport was able to assemble nuclear envelopes containing nuclear pores and a peripheral nuclear lamina around Lamda DNA, yet it remained unclear as to the lamina’s role in nuclear envelope assembly [36]. Immuno-depleting B-type lamin LIII (or B3) from nuclear assembly extracts did not prevent formation of a nuclear envelope consisting of both membranes and nuclear pores, although its integrity was negatively affected [85]. At this time lamin B3 was thought to be the only component of the lamina in oocytes and early embryos [86], suggesting that nuclear assembly was independent of lamins. It was only a few years later that another B-type lamin, lamin LII (or B2) was discovered and shown to play a role in nuclear assembly in early stage *Xenopus* embryos [37]. Following this work, Lourim and Khrone proposed a model for lamin-dependent nuclear envelope assembly [39]. According to this model, lamin-associated vesicular nuclear envelope precursors would bind to the surfaces of chromosomes and fuse to form a continuous membrane around the chromatin, which was followed by import of soluble lamins to assemble the nuclear lamina. Consistent with this model, it has been shown that beads coated with DNA are sufficient to attract nuclear assembly components and induce the formation of functional nuclear envelopes in *Xenopus* egg interphase extract [87]. The discovery of a third B-type lamin, lamin LI (or B1) [38], suggested that the three different lamins are involved in the formation of the nuclear envelope in nuclear reconstitution egg extracts. The three types of lamins were shown to associate with three independent vesicle populations and each of them were able to independently assemble nuclei in extracts. Of note, while their ratios vary as a consequence of changes in protein synthesis, the lamin B3 remains the most abundant during meiotic maturation [38]. The idea that the nuclear envelope reforms from vesicles was later challenged by live-cell imaging, where it was revealed that in vivo, the endoplasmic reticulum network remains intact during cell division [88]. Furthermore, the presence of endoplasmic reticulum vesicles in egg extracts was shown to actually result from reticulum network fragmentation which occurs during egg fractionation [89]. By generating interphase extracts with an intact endoplasmic reticulum, the authors were able to show that the nuclear envelope reforms from endoplasmic reticulum tubules in the absence of membrane fusion, which was concomitantly demonstrated in vivo [90,91]. Central to the proper function of nuclear envelopes are the nuclear pores, whose assembly also relies on the lamina. Indeed, aberrant positioning of nuclear pore baskets in nuclear envelope was observed in nuclei lacking a lamina [92]. While identifying Nup153 (a type of nucleoporin, proteins composing the nuclear pores) as being specifically associated with lamin B3, Smythe et al. found that this nucleoporin was incorporated into nuclear pores during lamina assembly [41]. Interestingly, preventing lamina assembly altered the localization of Nup153 at the nuclear envelope but did not affect the distribution and recruitment of other nucleoporins.

Proper assembly of the nuclear lamina is not only important for nuclear envelope formation, but it is also involved in DNA replication. Indeed, nuclei assembled in extracts functionally depleted of lamin B3 were unable to initiate DNA replication, despite nuclear importation of the proteins needed for this process [93]. Through depletion and add-back of purified lamin, it was further demonstrated that lamin is required for the assembly of a nucleus capable of replication [92]. Furthermore, addition of a truncated human lamin as a dominant-negative mutant in *X. laevis* interphase egg extracts resulted in the total disruption of the lamina as well as a major decrease in DNA replication. Despite this, the nuclear distribution of proteins required for the initiation of replication, such as DNA polymerase α, remained unaltered. In contrast, the distribution of proteins involved in the elongation phase of DNA replication, like PCNA (Proliferating Cell Nuclear Antigen), was indeed altered, as the protein was found to localize to aberrant lamin aggregates in the nucleus. This suggested a regulatory role for nuclear lamins, not in initiation, but rather in elongation [42]. Comparing the effects of a dominant-negative lamin mutant with those of AraC, a reversible inhibitor that blocks replication at the onset of the elongation phase, and CIP, an inhibitor of the cyclin-dependent kinase (cdk) 2-dependent step of initiation, Moir and colleagues concluded that proper lamin organization is important for elongation but not for initiation [43]. Of note, preventing lamin polymerization using a peptide of the C-terminal domain of *Xenopus* lamin B3 resulted in inhibition of both chromatin decondensation and formation of the nuclear envelope [44]. The authors proposed that lamin polymerization is needed for both chromatin decondensation and binding of nuclear membrane precursors. Studies of the lamin-associated protein LAP2 (Lamin Associated Polypeptide 2) in *Xenopus* egg extracts revealed that it is a downstream effector of lamina assembly and allows membrane-chromatin attachment, which suggests that it acts as a promoter of DNA replication by influencing chromatin structure [94]. Interestingly, LAP2 localization was shown to depend on TPX2, which seems to be required for nuclear formation after mitosis [95].

Lamin polymerization is also involved in the regulation of nuclear size. Though lamin-depletion studies in extracts had established a role for the lamina in the regulation of DNA replication, it was also noted in 1995 that nuclei diameters were greatly reduced under the same experimental conditions [92]. It was subsequently shown that importin α binds to lamin B3 via its NLS (Nuclear Localization Signal) domain to maintain its solubility and promote interactions with lamin-binding proteins [96]. By comparing egg extracts from *X. laevis* and *X. tropicalis*, it was discovered that modulation of lamin B3 import rates through changes in importin α and Ntf2 concentrations were sufficient to recapitulate the difference in nuclear size observed between these two species, revealing a role for lamin proteins in nuclear size regulation [45]. Edens and colleagues then suggested that nuclear size is maintained by a balance between nuclear import-mediated nuclear growth and a regulation of lamina dynamics by PKC (Protein Kinase C) activity and localization. PKC indeed phosphorylates lamin B3, thus leading to its removal from the nuclear envelope and to a subsequent decrease in nuclear size [46]. Changing the concentrations of the different types of lamins in *Xenopus* egg extracts, it was found that low and high concentrations increased and decreased the nuclear size, respectively. Of note, total lamin concentration rather than specific lamin type, was found to be more important in controlling the size of nuclei [97]. Finally, targeting of importin α to the outer boundary of extract droplets prepared using physiological lipid extract, modulated the nuclear import of lamins and recapitulated nuclear scaling relationships observed during embryogenesis. Moreover, these relationships were altered by inhibitors that shift levels of importin α palmitoylation, and thus its ability to bind membranes [81].

## 7. The Roles of Nuclear Actin from Structuring the Nucleus to Regulating Chromatin States

Nuclear actin was observed over 50 years ago, and evidence for its multiple functions, ranging from the regulation of RNA polymerases, transcription factors and chromatin remodeling complexes to DNA damage repair, has accumulated (for review [98,99]). In *Xenopus*, a notable study revealed that intranuclear injection of anti-actin antibodies into oocyte nuclei blocked chromosome condensation, while injection into the cytoplasm had no effect [100]. In these cells, nuclear actin has been shown to provide structural rigidity to their relatively large nuclei [101], and also to play a role in stabilizing ribonucleoprotein droplets against gravity [102]. Modifications of egg extract preparations, including omission of cytochalasin [103] or using a weaker actin-perturbing drug [104], were used to investigate nuclear actin (Figure 4, (3)–(5)). Using extracts modified to recapitulate the accumulation of nuclear actin seen in embryos at the blastula stage, it was shown that F-actin accumulates at the subnuclear membranous region and functions to maintain the binding of chromatin to the nuclear envelope. Moreover, nuclear actin has been implicated in the stiffening of nuclear lamina, increasing the stability of nucleus and contributing to chromosome alignment during spindle assembly [103]. In addition, it was recently demonstrated that actin binding to RanGTP-importin complexes is important for importin-cargo release, and that the actin elongating factor formin promotes loading and activation of DNA replication factors and thus initiation and elongation stages [104].

The results of immuno-depletion/add-back experiments in extracts in which the levels of the actin-interacting protein 4.1 (protein 4.1) were modulated, suggested a role for the protein in nuclear assembly. While both its spectrin-actin-binding domain (SABD) and a C-terminal domain that interacts with NuMA were found to be critical for nuclear assembly, the interaction between the protein’s SABD and actin seemed to be selectively important for nuclear assembly [28]. Further investigation showed that protein 4.1 is associated with actin in nuclei. Interestingly, inhibition of either protein 4.1 function using its SABD as a dominant negative or actin polymerization prevented the incorporation of both protein 4.1 and actin within nuclei. Finally, addition of the F-actin depolymerizer latrunculin A leads to small and irregular nuclear structures with few pores and lamins which are unable to initiate DNA replication, a phenotype not observed in extracts only treated with cytochalasin [29], further implicating nuclear actin as playing an important role in nuclear assembly and function.

## 8. The Control of Nuclear Assembly and Disassembly by the Microtubule Cytoskeleton

While microtubule function, from what has been elucidated so far, seems restricted to the cytoplasm, these filaments have been implicated in the assembly and disassembly of the nucleus. Some of the mechanistic underpinnings linking microtubules and nuclear growth dynamics have been revealed using *Xenopus* egg extracts (Figure 4, (6–8)). Notably, perturbations of microtubule dynamics and motor activity have demonstrated that polymerized microtubules are needed to properly assemble the nuclear envelope [27,105]. Indeed, addition of nocodazole, colcemid, as well as AS-2 (inhibitor of kinesin-mediated microtubule-dependent transport) did not prevent the formation of the nuclear envelope, but nuclei formed under these conditions lacked nuclear pore complexes and were therefore unable to accumulate karyophilic proteins. It is interesting to note that the assembly of annulate lamellae, a cytoplasmic structure forming stacks of membranes containing pore complexes similar to those of nuclear envelope but different in protein composition (studied in more details in mammalian cells [106,107]), was not affected by these same perturbations [105]. In addition, by combining *Xenopus* egg extracts with microfluidic devices to control the space occupied by microtubules around nuclei, Hara and Merten showed that dynein-based accumulation of membranes regulates nuclear expansion [27]. These two studies suggest that different mechanisms are involved in the assembly of the nuclear envelope—some which depend on microtubules and others which do not. Furthermore, microtubules also seem to be responsible for regulating the shape of the nucleus. For example, depleting Dppa2 (Developmental pluripotency-associated 2), a chromatin binding protein, which also regulates microtubule dynamics, resulted in increased microtubule polymerization, aberrant nuclear morphology, and defects in DNA replication. Strikingly, ectopic depolymerization of microtubules prior to nuclear assembly was sufficient to reverse these nuclear defects. This demonstrated the importance of spatiotemporal regulation of microtubules through local inhibition of polymerization during early nuclear formation by chromatin-binding microtubule regulators [26]. Of note, *Xenopus* egg extract was instrumental to screen for microtubule-membrane linkers and identify REEP4, a previously uncharacterized endoplasmic reticulum protein, which, together with REEP3, was shown in cell culture to ensure endoplasmic reticulum clearance from metaphase chromatin and proper nuclear envelope architecture [108]. Interestingly, depletion of γ-tubulin from egg extracts, which has been widely implicated in microtubule nucleation, resulted in disruptions in nuclear envelope and lamina assembly. Further experiments done in mammalian cells revealed that it is able to build its own meshwork, so-called γ-strings. Nuclear γ-strings seem to form around chromosomes and connect to both lamins and cytosolic γ-strings [109].

Finally, in addition to promoting proper nuclear envelope assembly, microtubules appear to be involved in nuclear envelope breakdown (NEBD). Investigating the role of microtubules and RanGTP in NEBD, Mühlhäusser and Kutay showed that the RanGTP gradient around chromatin may serve as spatial cue for NEBD and that microtubules were required for the completion of NEBD by participating in the removal of nuclear membranes from the vicinity of chromatin [25].

## 9. Conclusions

As our understanding of cell physiology evolves to integrate both biophysical and biochemical views of important biological molecules and their higher order assemblies, systems like *X. laevis* egg extracts will become increasingly useful. With the advent of microfluidics and micro-patterning, researchers have been able to use *Xenopus* egg extracts to assess the impacts of boundaries and cytoplasmic volume on self-organization at cellular length scales, in a way amenable to quantitative microscopy and imaging. *Xenopus* egg extracts provide a more accurate physiological context, which is often lost in traditional in vitro studies, while offering experimental flexibility in the geometry and volume of the discretized egg extract not found within in vivo models. In addition, the open nature of the cell-free extracts has allowed for molecular perturbations that may otherwise prove difficult in competing systems, providing a platform to disentangle the contributions and interactions between key proteins involved in nuclear assembly and the maintenance of cytoskeletal networks among other features of cell physiology. Importantly, *Xenopus* egg extracts have been instrumental in characterizing the self-organization of the cytoplasm, and more specifically, the cytoskeletal elements responsible for self-assembling and facilitating the coordination and movement of membranous structures within the cell. The degree to which these cytoskeletal elements interact with one another and the rest of the cytoplasm is poorly understood, as is the extent to which cells rely on this organization for crucial events such as mitosis and cell-motility, and more generally, their homeostasis. As methods for the depletion of target proteins from *Xenopus* egg extract advance, and perhaps more vitally, the imaging of endogenous proteins at a higher temporal and spatial resolution, so too will our understanding of these emergent cytoskeletal processes and their importance.

## Figures and Tables

**Figure 1 cells-10-02197-f001:**
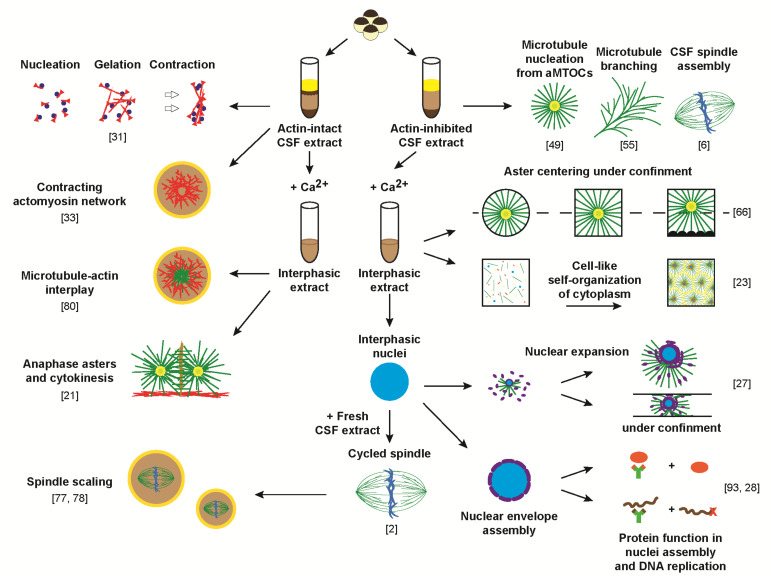
Reconstitution of cytoskeleton-dependent self-organization processes using the *Xenopus* egg extract in vitro system. Studies introducing or utilizing the reconstitution system in the context of investigating self-organization are referenced next to corresponding schematics in the order they appear in the text.

**Figure 2 cells-10-02197-f002:**
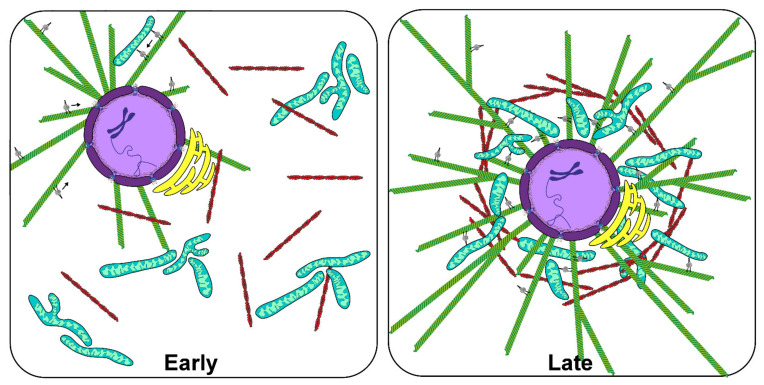
Microtubule-dependent self-organization of *Xenopus laevis* egg extract. Representative schematics of early and late stage organization of *X. laevis* egg extract during interphase. In the early stage (left panel), a microtubule aster (green) associated with the nucleus (purple) begins to expand and makes contact with dispersed elements of the cytosol including the ER (yellow), mitochondria (blue), and actin cytoskeleton (red). After reaching steady state (right panel) the microtubule cytoskeleton has centered itself in the region of interest, together with it the associated nucleus and ER. Clustering of mitochondria towards the “cell” center through the activities of minus-end directed motors such as cytoplasmic dynein (gray) occurs concurrently with microtubule aster centering and continues until most membranous cargos are cleared from more distal regions.

**Figure 3 cells-10-02197-f003:**
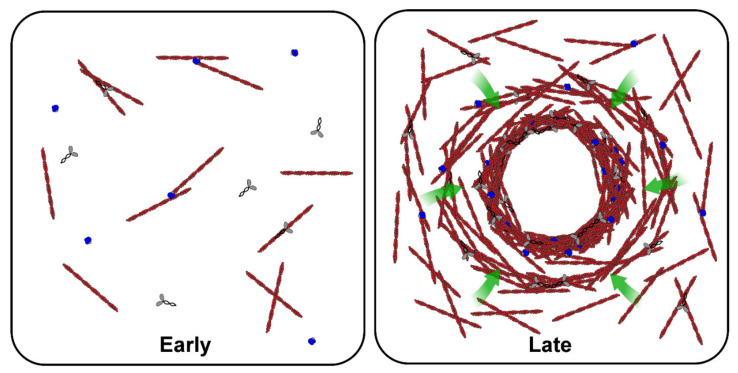
Gelation-contraction of the actomyosin network. Representative cartoon showing the distributions and rearrangements of F-actin (red), actin nucleators (blue), and myosin (gray) during early and late stages of actomyosin contraction in interphase *X. laevis* egg extracts. Shortly after F-actin nucleation (early stage; left panel), the F-actin network undergoes contraction (late stage; right panel), resulting in cytosolic flows towards the aggregate center (green arrows) and an F-actin density gradient.

**Figure 4 cells-10-02197-f004:**
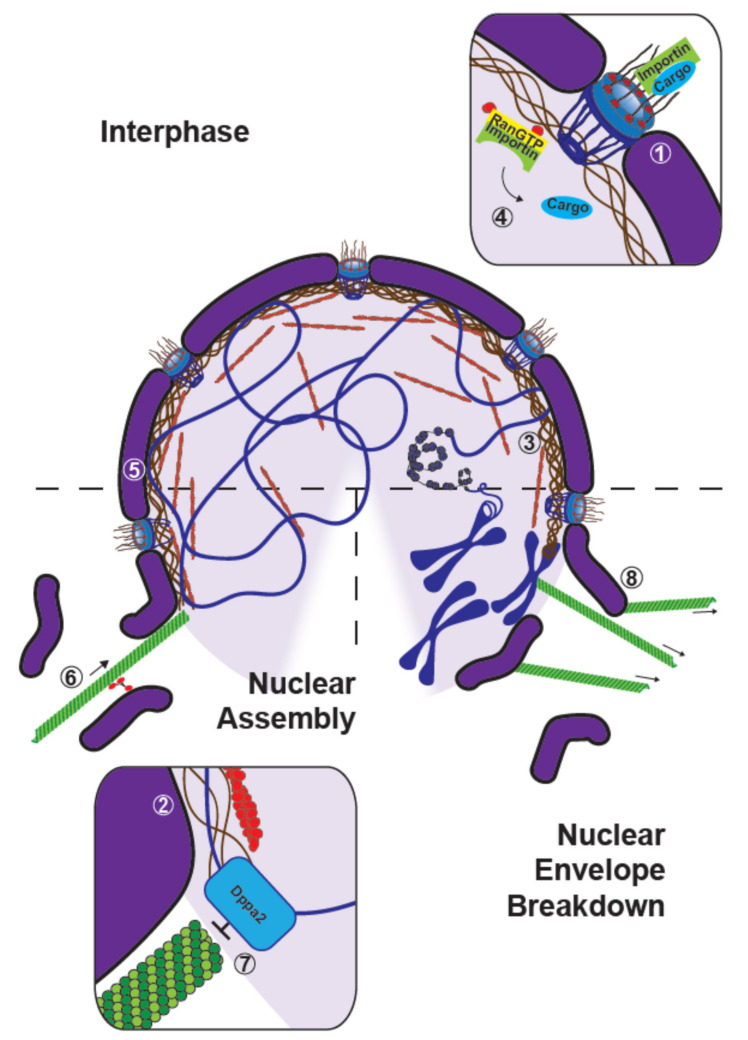
The cytoskeleton and nuclear assembly, structure, and breakdown. Overview schematics of the different contributions of the cytoskeleton to the formation and maintenance of the nucleus. Nuclear envelope is shown in purple, chromatin in blue, lamin filaments in brown, actin in red and microtubules in green. (**1**) Lamin B3 promotes the recruitment of the nucleoporin, Nup153, into nuclear pores and to the nuclear envelope. (**2**) The Lamina-associated protein LAP2 contributes to DNA-nuclear membrane interactions, and ultimately to the elongation of DNA during replication. (**3**) F-actin accumulates at the sub-nuclear membranous region to stiffen the nuclear lamina and maintain the binding of chromatin to the nuclear envelope. (**4**) Nuclear actin helps the release of cargoes from RanGTP-importins. (**5**) The actin-interacting protein, protein 4.1, relies on its spectrin-actin-binding domain (SABD) to interact with nuclear actin and contribute to the nuclear formation. (**6**) Minus-end directed motors transport membranous nuclear envelope precursors to assemble the nucleus. (**7**) The chromatin binding protein, Dppa2, inhibits microtubule polymerization near chromatin to allow for proper nuclear assembly. (**8**) Microtubules may serve a role in removing membranes during nuclear envelope breakdown.

## Data Availability

Not applicable.

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
