# Peer review of "The Cytoskeleton and Its Roles in Self-Organization Phenomena: Insights from Xenopus Egg Extracts"

_cells, 2021, doi:10.3390/cells10092197_

Round 1

Reviewer 1 Report

The Xenopus extract system is widely known as a powerful in vitro system to understand cell cycle control, DNA replication and spindle assembly. However, it has many advantages for the study of cytoplasmic organization. In this review, the authors do an outstanding job of assembling an extensive and important body of literature focused on mechanisms controlling organization of the actin and microtubule cytoskeletons and the nuclear lamina. While the other aspects of the Xenopus system have been reviewed extensively, to my knowledge the current manuscript is the first to focus exclusively on cytoskeletal organization in this system.  It therefore will be an important contribution to anyone interested in the cytoskeleton, as well as to scientists who are interested in the more conventional aspects of the Xenopus system. In addition, it has the potential to stimulate ideas on other novel ways that this powerful extract system could be utilized. 

Overall, the manuscript is well-written and puts the important findings in the historical context of their discovery while connecting these findings to current models of cytoskeletal organization.  I am in strong support of publication after a fresh proofreading to fix a few minor grammatical errors. 

Author Response

We thank the reviewer for the positive feedback. We performed a fresh proofreading as requested and fixed a few mistakes and grammatical errors.

Reviewer 2 Report

The review “The cytoskeleton and its roles in self-organization phenomena: insights from Xenopus egg extracts” by Geisterfer and colleagues is a really interesting and exhaustive focus on the use of Xenopus eggs extract in our current understanding of cytoskeletal self-organization. The review uses a nice and original historical narrative thread in the introduction and highlight the advantages and the value of this system for many aspects of our understanding of cell biology and self-organization of and by cytoskeletal filaments. 
The manuscript is well written and illustrated by clear Figures. The manuscript nicely shed light on this great system and will definitely be of interest for many researchers in the respective fields covered by this review.  

Here are few comments that could improve the manuscript: 
When mentioning the actin-microtubule interplay, I would have include the work from Colin et al. 2018 DOI:https://doi.org/10.1016/j.cub.2018.06.028 where authors show clear changes in microtubule network dynamics and organization induced by different F-actin architectures. This reference seams totally relevant for the points addressed in this manuscript.

About Figures:
Figure 1 is really useful and represent by itself an exhaustive summary of the review and of the great value of Xenopus eggs extract in our understand of sub-cellular cellular self organization. Even if the references are mentioned all along the manuscript, I would however, cite references on the Figure itself or at least in the legend, especially since Refs format in Cells [numbered] will not overload the figure or legend.
Figure 4 is quite clear but might need legends of all different filaments and features depicted, especially for readers that are not familiar with nuclear envelope organization. 

Author Response

We are grateful to the reviewer for the positive feedback and useful suggestions.

The work from Colin et al. 2018 is now mentioned and cited page 9, lines 339-341.

Figure 1 has been revised so that it now includes the reference numbers of relevant papers.

Figure 4 has been revised as suggested by reviewer 3 and the depicted features are now identified in the legend as requested.

Reviewer 3 Report

Geisterfer and colleagues prepared overview the cytoskeleton and its roles in self-organization based on research with Xenopus egg extracts. Authors review the understanding of the cytoplasmic aspects of self-organization by the microtubule and the actomyosin cytoskeletons as well as the importance of cytoskeletal filaments in organizing nuclear structure and function.

The paper is well written and its current form is almost sufficient for publication in Cells.

Minor point which I'd like to be adress is small modification on Figure 4. Magnifications are almost the same size as whole picture. My advice is to shrink whole figure as it is and prepared bigger magnification so reader could easly see subtitels (cargo, importin etc. should be visable as good as now after magnification document til 200%).

Author Response

We thank the reviewer for the feedback and for supporting the publication of our manuscript in Cells. We revised Figure 4 as requested and hope this will be satisfactory to the reviewer.